# Consequences of Adhesion Molecule Close Homolog of L1 Deficiency for Neurons and Glial Cells in the Mouse Spinal Cord After Injury

**DOI:** 10.3390/biom15091247

**Published:** 2025-08-28

**Authors:** Igor Jakovcevski, Ayse Acar, Benjamin Schwindenhammer, Mohammad I. K. Hamad, Gebhard Reiss, Eckart Förster, Melitta Schachner

**Affiliations:** 1Institut für Anatomie und Klinische Morphologie, Universität Witten/Herdecke, 58455 Witten, Germany; a.acar@web.de (A.A.); benjamin.schwindenhammer@uni-wh.de (B.S.); gebhard.reiss@uni-wh.de (G.R.); 2Department of Neuroanatomy and Molecular Brain Research, Institute of Anatomy, Ruhr-Universität Bochum, 44780 Bochum, Germany; eckart.foerster@rub.de; 3Department of Anatomy, College of Medicine and Health Sciences, United Arab Emirates University, Al Ain P.O. Box 15551, United Arab Emirates; m.hamad@uaeu.ac.ae; 4W. M. Keck Center for Collaborative Neuroscience, Department of Cell Biology and Neuroscience, Rutgers University, Piscataway, NJ 08854, USA

**Keywords:** close homolog of L1, glia, motoneurons, neurons, interneurons, parvalbumin, spinal cord injury

## Abstract

After spinal cord injury, pathological changes predominantly proceed caudal to the site of injury. To what extent these changes contribute to abnormalities during regeneration is poorly understood. Here, we addressed this question with a low-thoracic compression injury mouse model. The total numbers of immunohistochemically stained neuronal and glial cell types in the lumbar spinal cord were stereologically determined 6 weeks after injury. We also investigated injured mice deficient in close homolog of L1 (CHL1), which had been reported to recover better after injury than their wild-type littermates. We here report that there were no differences between genotypes in uninjured animals. In both injured CHL1-deficient and wild-type littermates, gray and white matter volumes were decreased as compared with uninjured mice. Numbers of motoneurons and parvalbumin-expressing interneurons were also reduced in both genotypes. Numbers of interneurons in injured mutant mice were lower than in wild-type littermates. Whereas injury did not affect numbers of astrocytes and oligodendrocytes in the gray matter, numbers of microglia/macrophages were increased. In the mutant white matter, numbers of oligodendrocytes were reduced, with no changes in numbers of astrocytes and microglia. A loss of motoneurons and interneurons was observed in both genotypes, but loss of interneurons was more prominent in the absence of CHL1. We propose that, after injury, CHL1 deficiency causes deficits in structural outcome not seen after injury of wild-type mice.

## 1. Introduction

The limited regeneration potential of the adult mammalian spinal cord is a serious challenge for biomedical science. The lack of regenerative capacity is caused by the injury-induced degeneration of neurons on the one hand and by changes in the environment at the lesion site on the other [1,2,3]. Considerable numbers of various cell types, including neurons, astrocytes, oligodendrocytes, and their precursor cells, die at the injury site. Vascular trauma caused by mechanical forces entails disruption of the blood–brain barrier, and disturbed blood circulation leads to hypoxia. Parenchymal and vascular damage leads to secondary loss of neurons and glial cells, accompanied by activation of inflammatory and wound-healing responses [4]. Inflammatory cells have a range of both destructive and reparative roles after spinal cord injury, many of which are still poorly understood [5,6]. Whereas the cellular composition, the molecular expression patterns, and the progression of the glial scar at and closely around the site of injury are well characterized, there is only sparse and largely controversial data on the histological alterations in the spinal cord caudal to the lesion site. Lumbar spinal cord segments contain motoneurons, which provide axons to spinal nerves that are involved in hindlimb control. Axonal regrowth and sprouting in these segments were observed earlier, showing that only a few axons regenerate over long distances to reach their original targets [3,7]. 

Among the cell adhesion molecules of interest for regeneration is the close homolog of adhesion molecule L1 (CHL1), shown to be upregulated by astrocytes at the lesion site after spinal cord injury [8]. This upregulation of CHL1 expression was considered to hamper regeneration through homophilic CHL1-CHL1 interactions between axons and astrocytes, leading to better post-injury recovery in female, but not male CHL1−/− mice [8,9]. Previously, we had focused on the injury site rather than the lumbar spinal cord, which contains motoneurons and interneurons that are responsible for the innervation of hindlimb muscles. Based on these observations, we now used stereology to count all major cell populations in the lumbar spinal cord after low-thoracic injury in CHL1−/− mice and their wild-type (CHL1+/+) littermates, using the same experimental protocol as before [8]. In this present study, we used only female mice, since it was not possible to obtain approval for injured male mice because bladder voiding after injury is more painful for males than for females.

Different observations had been reported regarding the changes in neuronal cell populations in the rodent lumbar spinal cord after thoracic injury [10,11,12,13]. The use of different animals, such as cats, rats, or monkeys, as well as different methods of injury, are likely the reason for these differences [10,11,12,13,14,15]. The different, sometimes biased methods for assessment of cell numbers may have contributed to controversies. Here, we used a quantitative, i.e., stereological, approach for immunohistochemical visualization of defined cell types in the hope of gaining novel insights into the pathophysiology of mouse spinal cord injury. Stereology allows for concurrent analysis of defined neural cell populations [6]. Furthermore, we also analyzed CHL1−/− mice as an example of a genotype that unexpectedly recovers better after injury compared to CHL1+/+ littermates [8]. The present stereological analyses were focused on astrocytes, oligodendrocytes, motoneurons, and microglia-macrophages, representing the major cell types in the lumbar spinal cord before and after injury.

## 2. Materials and Methods

### 2.1. Animals 

Altogether, 12 CHL1−/− mice and 12 of their CHL1+/+ wild-type littermates on the C57BL/6J genetic background [16] were obtained from the heterozygous breeding colony at the Center for Neurobiology Hamburg. Animals were 12 weeks old. They were maintained under standard laboratory conditions. In total, 6 CHL1+/+ and 6 CHL1−/− mice were subjected to spinal cord injury, and 6 CHL1+/+ and 6 CHL1−/− mice were non-injured controls. Tissue samples were analyzed from non-injured mice (further “uninjured” or “without injury”) and from injured mice at 6 weeks after injury (further “injured” or “after injury”). All experiments were conducted in accordance with the German and European Community laws on the protection of experimental animals, and the procedures used were approved by the responsible committee of the State of Hamburg (permit numbers 18/07 and 98/09, date 14 December 2009). Numbers of animals used for experiments are given in the figure legends. All animal treatments, data acquisition, and analyses were performed in a blinded fashion, according to the ARRIVE guidelines, which were also followed for preparing the manuscript. 

### 2.2. Surgical Procedures 

Mice were anesthetized by intraperitoneal injection of ketamine and xylazine (100 mg Ketanest^®^, Parke-Davis/Pfizer, Karlsruhe, Germany, and 5 mg Rompun^®^, Bayer, Leverkusen, Germany, per kg body weight). Laminectomy was performed at the T7-T9 levels with a mouse laminectomy forceps (Fine Science Tools, Heidelberg, Germany). A mouse spinal cord compression apparatus was used for injury [8]. Compression force (degree of closure of the forceps) and duration were controlled by an electromagnetic device. The spinal cord was maximally compressed for 1 second by a time-controlled 12 V current flow through the electromagnetic device. The skin was then surgically closed with 6-0 nylon stitches (Ethicon, Norderstedt, Germany). Mice were then allowed to recover in a heated room (35 °C) for several hours to prevent hypothermia. Thereafter, mice were singly housed in a temperature-controlled (22 °C) room with water and standard food ad libitum. Bladders were manually voided twice daily.

### 2.3. Antibodies

The following commercially available antibodies were used for immunohistochemistry at dilutions as determined [6,8]: anti-parvalbumin (PV, mouse monoclonal, clone PARV-19, catalog number P3088, Sigma, Deisenhofen, Germany, 1:1000), anti-neuron specific nuclear antigen (NeuN, mouse monoclonal, clone A60, catalog number MAB377, Sigma, 1:1000), anti-choline-acetyl transferase (ChAT, goat polyclonal, catalog number AB144P, Sigma, 1:100), rabbit anti-ionized calcium binding adaptor molecule 1 (Iba1, catalog number 019-19741, Wako Chemicals, Neuss, Germany, 1:1500), rabbit anti-S100b (catalog number GA504, DakoCytomation, Glostrup, Denmark; 1:2000), 2′,3′-cyclic nucleotide 3′phosphodiesterase (CNP; clone 2872 mouse monoclonal, catalog number AMAB91069, Sigma; 1:1000).

### 2.4. Motor Function Analysis

Before injury (further referred to as time-point 0 days) and 1, 3, and 6 weeks after injury, locomotor function of mice was evaluated by the Basso Mouse Scale (BMS) score, and videos of mice in the beam-walking test were recorded to calculate foot-stepping angle (FSA) [6,8]. Briefly, in the BMS score, mice were allowed to walk in an open field arena, and both lower limbs were evaluated according to predefined parameters [6]. Videos of the beam-walking test were recorded on a 1 m-long wooden beam, 10 cm wide, after the mouse had been pre-trained 3–4 times to run the beam towards the home-cage without stopping. The angle between the beam and foot of a mouse was measured using Fiji freeware [6,8]. Videos were taken from both sides, and values for both lower limbs were averaged.

### 2.5. Tissue Fixation and Sectioning 

Mice were weighed and anesthetized with a 16% solution of sodium pentobarbital (Narcoren, Merial, Hallbergmoos, Germany) at 5 µL/g body weight, i.p.). After surgical tolerance was achieved, as checked by tweezer tail pinch, the mice were transcardially perfused for 60 seconds with saline followed by the fixative, consisting of 4% formaldehyde and 0.1% CaCl_2_ in 0.1 M cacodylate buffer, pH 7.3, for 15 min at room temperature (RT). Thereafter, the spinal cords were left in situ for 2 h at RT and removed for post-fixation overnight (18–20 h) at 4 °C in the perfusion solution. The tissue was then immersed for two days in 15% sucrose solution in 0.1 M cacodylate buffer, pH 7.3, at 4 °C.

Fixed and sucrose-infiltrated cryoprotectant spinal cords were examined under a stereomicroscope (2× zoom, ZEISS Stemi 305, Zeiss AG, Oberkochen, Germany), and remaining tissue pieces of dura mater were removed with the forceps. Afterward, the spinal cords were placed in 1 cm-long thin foil chambers filled with Tissue Tek (Satura Finetek Europe, Zoeterwoude, The Netherlands). Finally, the spinal cords were frozen by insertion into 2-methyl-butane precooled in a –80 °C freezer for 2 min. The spinal cords were stored at –80 °C. The part of the lumbar spinal cord caudal to the site of injury was sectioned on the cryostat Leica CM3050 (Leica Instruments, Nußloch, Germany), with the rostral end facing the cryostat blade. Sections (25 μm thick) were collected on Super Frost Plus glass slides (Roth, Karlsruhe, Germany). Because stereological analyses require spaced serial sections [17], sampling was performed in a sequence such that six sections 250 µm apart were present on each slide.

### 2.6. Immunohistochemistry

Immunohistochemical stainings were performed by indirect immunofluorescence as described [16]. Twenty-five-μm-thick sections were air-dried for 30 min at RT. They were then incubated at 80 °C with antigen unmasking solution (10 mM sodium citrate, pH 9.0) for 30 min and cooled down to RT. Afterward, blocking of unspecific binding sites was performed at RT for one hour in the blocking solution, PBS solution of 0.2% Triton X-100 (Fluka, Buchs, Germany), 0.02% sodium azide (Merck, Darmstadt, Germany), and 5% non-immune goat or donkey serum (Jackson Immunoresearch Laboratories, Dianova, Hamburg, Germany), the selection of which was determined by the species that generated the secondary antibody, for 1 h. The slices were afterward incubated with the primary antibody diluted in PBS containing 0.5% lambda-carrageenan and 0.02% *w*/*v* sodium azide in PBS. To ensure sufficient antibody penetration through the tissue section, the slides were incubated for 3 days at 4 °C in a staining jar. On the fourth day of staining, sections were rinsed 3 times in PBS (15 min each) before the appropriate secondary antibody was incubated at RT for 2 h at a dilution of 1:200 in PBS-carrageenan solution. We used goat anti-rabbit, goat anti-mouse, and donkey anti-goat IgG antibodies conjugated with Cy3 (Jackson Immunoresearch Laboratories, Dianova, Hamburg, Germany). Then, sections were washed in PBS, and cell nuclei were stained for 10 min at RT with 4,6-diamidino-2-phenylindole solution (DAPI, Merck, Darmstadt, Germany). After another wash, sections were mounted in Fluoromount G (Southern Biotechnology Associates, Biozol, Eching, Germany), coverslipped, and stored in the dark at 4 °C until use.

### 2.7. Stereological Analysis 

The optical dissector method was chosen for quantification of numerical densities of different cell types in a given tissue volume [16]. The method consists of the direct counting of objects in 25 µm thick sections using a three-dimensional counting frame (“counting brick”, see [17], referred to as “dissector”). The base of the frame, defined by the size of the grid squares, and the height of the dissector, defined as a portion of the section thickness in the z-axis, were controlled by the electronic device [17]. The cells were counted on an Axioscope microscope (Zeiss) equipped with a motorized stage and a Neurolucida software-controlled computer system (MicroBrightField, Colchester, VT, USA). Sections were firstly observed under 10× objective with a 365/420 nm excitation/emission filter set (01, Zeiss, blue fluorescence). The nuclear staining by DAPI allowed delineating spinal cord structures and areas. The viewed area was randomized by arbitrary setting a reference point, resulting in an overlay grid with lines spaced at 60 µm (30 µm for NeuN and nuclear count). Squares within the marked area were chosen for counting, starting from the uppermost left side of the delineated field at distances of 60 µm apart for the other squares. A depth of 2–10 µm from the cut surface was chosen for visualization of stained structures. The sections were then viewed with a 40× objective and 546/590 nm excitation/emission filter set 15, red fluorescence (Zeiss). Per animal and staining, six sections spaced 250 μm apart were evaluated bilaterally. For PV-positive and ChAT-positive neurons, all immunopositive neurons in the ventral (for ChAT, PV) and dorsal (for PV) horns were counted together with nuclei within the dissector depth.

### 2.8. Estimation of Spinal Cord Volumes

The volume of the lumbar spinal cord was measured according to Cavalieri’s principle [17]. The outlines of the spinal cord boundaries in coronal sections were manually drawn under the microscope (4× objective) for determining the areas with Neurolucida software (Microbrightfield). The volume was calculated in mm^3^ by multiplying the sum of the areas per section by the distance between sections, i.e., 250 µm.

### 2.9. Photographic Documentation

Photographic documentation was performed on an LSM 510 confocal microscope (Zeiss) using a 63 × 2 oil immersion objective and digital resolution of 1024 × 1024 pixels. The images were processed using Adobe Photoshop CS5 software (Adobe System Inc., San Jose, CA, USA).

### 2.10. Statistical Analysis

Statistical analysis was performed using SigmaPlot 12 software (SYSTAT, Palo Alto, CA, USA) and GraphPad Prism 7 (Dotmatics, Boston, MA, USA). First, all data were analyzed for normal distributions using the Shapiro–Wilk and equal variance tests. As data were normally distributed and showed equal variances, we used parametric statistical tests, as indicated in the results and figure legends. Data were analyzed using one-way analysis of variance (ANOVA) followed by the Holm–Sidak post hoc test if ANOVA showed significant differences. Functional recovery was analyzed using two-way ANOVA for repeated measures, followed by the Holm–Sidak post hoc test. For all comparisons, the number of animals determined the degree of freedom. The accepted level of significance was 5%. Raw data (cell counts) generated in this study are available as Appendix A. 

## 3. Results

### 3.1. Reduction of the Lumbar Spinal Cord Volume Without Cell Loss After Injury

After injury, locomotor recovery and body weight were monitored weekly. As published earlier [8], at the sixth week after injury, recovery, demonstrated by the BMS score and foot-base angle, was better in CHL1−/− than in CHL1+/+ mice (Figure 1). Body weight measured before injury was similar in CHL1+/+ and CHL1−/− mice and declined in the first week after injury but later remained constant and similar in both genotypes.

After low-thoracic spinal cord injury, the most prominent morphological change in the lumbar spinal cord was the reduction of its volume. The volume was reduced by 30% in CHL1+/+ mice (25.4 ± 0.6 vs. 17.7 ± 0.8 mm^3^; p > 0.001, one-way ANOVA with Holm–Sidak post hoc test) and by 35% in CHL1−/− mice (25.3 ± 1.5 vs. 16.3 ± 0.6 mm^3^; *p* > 0.001, one-way ANOVA with Holm–Sidak post hoc test). The volumes of the uninjured lumbar spinal cords were the same in both genotypes. There were also no differences in volumes of injured spinal cords between CHL1+/+ and CHL1−/− mice. Volumes of the gray matter in the ventral horn, the dorsal horn, and the white matter decreased to a similar degree (Figure 1). This indicates that the loss of volume similarly affects the gray matter, which contains the neuronal cell bodies, and the white matter containing myelinated axons. Since there was a loss of volume, numerical data for cell density do not reflect the total cell number. Thus, we do not show data for cell densities, but rather total cell number, as calculated by multiplying cell density with volume. 

Numbers of DAPI-stained cell nuclei did not change when comparing injured versus uninjured animals. Also, there was no difference in cell numbers of CHL1+/+ versus CHL1−/− animals when comparing uninjured to injured mice (Figure 2). 

### 3.2. Loss of Motoneurons and Parvalbumin-Positive Interneurons After Injury

Regarding NeuN-expressing neurons in the ventral and dorsal horns, the ventral part of the spinal cord contains motoneuronal cell bodies, whereas the dorsal horn comprises mainly cell bodies of sensory neurons [18]. NeuN is observed in most neurons in uninjured and injured spinal cords, either in the ventral or in the dorsal horns, although in the ventral horn there was a tendency for a decrease after injury (Figure 3; *p* = 0.21). Also, genotypes of uninjured and injured mice did not show different numbers of neurons.

Numbers of motoneurons in the ventral horn identified by ChAT immunostaining were decreased in CHL1+/+ and –45.5% in CHL1−/− mice (–41.5% in CHL1+/+ and –45.5% in CHL1−/− mice), with no difference between genotypes (Figure 4).

Besides motoneurons, another interesting class of neurons are parvalbumin-positive interneurons [19,20,21]. In both ventral and dorsal horns, the numbers of these neurons were decreased after injury in CHL1+/+ mice (–25% and –38% for the ventral horn and for the dorsal horn, respectively). Furthermore, the number of interneurons was lower in CHL1−/− than in CHL1+/+ mice (–54% and –61% for the ventral horn and dorsal horn, respectively) (Figure 5). There was no difference between genotypes in uninjured mice, but a significant difference emerged after injury, suggesting that injury interacted with genotypes. Two-way ANOVA analysis revealed a significant interaction between factors “injury” and “genotype” (*p* = 0.018). Thus, although there was no general, overall loss of neurons after spinal cord injury, there was a considerable loss of motoneurons and interneurons. The loss of interneurons in the dorsal and ventral horns was more prominent in CHL1−/− than in CHL1+/+ mice.

### 3.3. Glial Cell Populations in the Lumbar Spinal Cord After Injury

The three major glial cell types are astrocytes, oligodendrocytes, and microglia/macrophages [6,22,23]. S100b+ astrocytes showed more prominent labeling in cell bodies than in processes (Figure 5A,B). There was no difference in the number of astrocytes in the ventral or in dorsal horns between uninjured and injured mice (Figure 5). Also, there was no difference between CHL1+/+ and CHL1−/− mice (Figure 6).

Anti-CNP antibody labels oligodendrocyte precursor cells and mature oligodendrocytes. Similar numbers of these cells were observed in the gray matter of uninjured and injured animals. Yet, in the white matter there was a decrease in the number of CNP+ cells in both CHL1+/+ and CHL1−/− mice (Figure 7). 

Numbers of Iba1-labeled microglia/macrophage cells were increased in the ventral horn after injury in both genotypes. In the dorsal horn and the white matter, the cell numbers tended to be higher, although not significantly, in uninjured versus injured animals (Figure 8).

## 4. Discussion

The aim of our study was to investigate the quantitative changes in cellular composition of the lumbar spinal cord after low-thoracic compression injury. The most prominent finding was the reduction in the lumbar spinal cord volume. This volume reduction is, however, not unexpected because axons degenerate, followed by demyelination. Furthermore, the numbers of motoneurons, parvalbumin-interneurons, and oligodendrocytes were decreased after injury. The only increase in cell number after injury was observed for microglia/macrophages in the gray matter. Results are summarized in Table 1. 

### 4.1. The Effect of Injury in CHL1+/+ Mice

In a complete mouse spinal cord transection model, a loss of approximately 20% of the ventral horn volume was reported upon injury, but no changes in the number of ventral horn neurons, leading to the conclusion that the volume loss was due to a reduction in neuropil [10]. Using an incomplete compression rather than a complete transection injury paradigm, we observed an even more prominent loss of volume in both gray and white matter. 

Spinal cord injury causes tissue destruction and total neuronal cell loss not only at the site of injury but also remote from the lesion. Since connections between higher centers and the lumbar spinal cord are disrupted, notably deafferentation affects spinal neurons. This deafferentation leads to anterograde transsynaptic degeneration of neurons, in particular of motoneurons, although the evidence for this view has been conflicting [10,11]. Anterograde transsynaptic degeneration after injury is caused by afferent denervation of neurons remote from the site of injury in several central nervous system injury situations and in neurodegenerative disorders [24,25]. In the spinal cord, loss of sensory input after peripheral nerve injury induces degeneration of spinal cord projections and interneurons in the dorsal horn [26]. Data about the loss of motoneurons after spinal cord injury in adult mammals are, however, conflicting. Some investigators reported a substantial reduction in the number of ventral horn neurons [11,27,28], whereas others have found no change in the number of motoneurons [10,13]. There are, however, obvious differences between these studies, concerning studied species, injury severity, and methods of counting. Most likely, the severity of injury is the most important factor, and this is exemplified by the degree of neurodegeneration, as suggested for human patients [29].

After injury, we also detected a considerable reduction in the number of parvalbumin-positive interneurons in both the ventral and the dorsal horns. When compared with uninjured mice, the total number of NeuN+ neurons had a tendency for a reduction of 22% in the ventral horn, while the tendency in the dorsal horn was smaller, but both were not statistically significant. Since we detected a loss of interneurons, but not a general loss of neurons after injury, our data indicate higher vulnerability of parvalbumin-positive interneurons compared to other types of neurons. Parvalbumin expression levels vary substantially between interneuron subpopulations in the intact spinal cord [19,30]. Importantly, in many other models of neurological diseases, selective vulnerability of parvalbumin-positive interneurons has been reported. For example, in animal models of epilepsy [31], Alzheimer’s disease [32], and interleukin 6 overexpression [33], interneurons were specifically affected, while other types of neurons were not. There is evidence that parvalbumin expression may protect neurons from glutamate-induced excitotoxic cell death. Transgenic mice ectopically overexpressing parvalbumin in motoneurons show decreased injury-induced motoneuronal cell death [34]. We propose that a reduced interneuron number after injury may reduce functional recovery. However, reduced inhibition from parvalbumin-expressing interneurons could counterintuitively increase locomotor function due to a higher spasticity in the lower limbs or cause additional, still unknown compensatory remodeling mechanisms in the spinal cord circuitry, which could lead to better locomotion. It is noteworthy that CHL1−/− female mice show better preservation and/or regeneration of cholinergic synaptic terminals around motoneurons [8].

Numbers of dorsal horn neurons in lumbar spinal cords did not reveal differences between injured and uninjured animals. The number of interneurons in the dorsal horn was reduced by about 50%, but the total number of NeuN+ neurons in the dorsal horn was unchanged after injury in CHL1+/+ mice. We can, thus, conclude that the number of dorsal horn projection neurons, which relay sensory information from dorsal root ganglia to higher brain centers, was not reduced by injury. This finding is puzzling because axons of these neurons are injured at the lesion site. Afferent input to these neurons is unaffected by spinal cord injury. It is well documented across species and injury models that, upon peripheral nerve injury, dorsal horn neurons undergo transsynaptic degeneration and death [26]. Even if the number of sensory spinal cord neurons is not affected by injury, their function is altered, since chronic pain after injury, mediated by dorsal horn projection neurons, is one of its most prominent symptoms after injury [35,36,37]. CHL1 might be of interest in the context of chronic pain, as its upregulation by dorsal horn neurons upon injury is implicated in neuropathic pain [38].

There are conflicting data on how widespread astrogliosis is after injury. An apparent increase in the density of astroglia after injury reported in many studies [4,5] is also observed in this present study, but this is due to a high reduction in the volume of spinal cord after injury caudal to the lesion site, but the number of astrocytes remains unchanged. There is, however, evidence that some features of astrocytes are affected by injury; for example, we previously detected elevated GFAP levels as measured by Western blot from tissue of the lumbar spinal cords of injured CHL1−/− and CHL1+/+ mice [8]. Regarding demyelination/remyelination events after spinal cord injury, we analyzed in the present study the spinal cord at a time-point when proliferation of oligodendrocytes had already ceased, while demyelination and remyelination may still proceed [39]. Thus, a decrease in oligodendrocyte density in the white matter 6 weeks after injury would indicate a deficit in myelination due to the severity of injury. No changes, compared to uninjured mice, were found in the number of gray matter oligodendrocytes in either genotype. The differential response of the oligodendrocytes in the white versus the gray matter may be explained by the possibility that oligodendrocytes in the gray matter are less susceptible to damage or have a higher proliferation rate [40]. Since the number of CNP-expressing oligodendrocytes was similar in both genotypes, we did not perform a more detailed analysis of remyelination after injury using myelin-specific markers such as myelin basic protein. We acknowledge that this is a weakness of our study. Microglia/macrophages are found at and remote from the injury site [41], as observed also in this present study. The increase in the numbers of microglia/macrophages remote from the lesion site could likely refer to their roles in eliminating neurons which undergo excitotoxic necrotic cell death, and synaptic remodeling [6,8]. The increase in the number of microglia/macrophages in the ventral horn could indicate more severe neurodegeneration in this region. This is corroborated by our finding of a large reduction in the numbers of motoneurons and parvalbumin+ interneurons in the ventral horn after injury, and, to a lesser degree, in the dorsal horn. It was also shown that the ablation of microglia/macrophages does not affect locomotor recovery [6].

### 4.2. The Effect of Injury in CHL1−/− Mice

Surprisingly, in contrast to the previous observations in the hippocampus and cerebellum of CHL1−/− mice, in uninjured animals there was no difference between CHL1+/+ and CHL1−/− mice in any major cell population [16]. There was no difference in astrocyte, oligodendrocyte, or microglial/macrophage numbers between genotypes, indicating that CHL1 does not affect glial proliferation after injury. 

The upregulation of CHL1 expression by astrocytes at the injury site has an adverse effect on regeneration [8]. The number of astrocytes in the lumbar spinal cord after injury was not different between CHL1−/− and their CHL1+/+ littermates, although GFAP protein levels were higher in spinal cords of CHL1+/+ compared with CHL1−/− mice after injury [8]. Also, the numbers of Iba1+ cells, oligodendrocytes, motoneurons, and neurons were similar between the genotypes, with the exception that the number of interneurons was more reduced upon injury in CHL1−/− than in CHL1+/+ mice. This could be explained by a higher vulnerability of interneurons in the absence of CHL1 in the aging hippocampus of CHL1−/− mice [16]. This difference, however, does not appear to lead to functional consequences, since, unexpectedly, the recovery is better in female CHL1−/− than in CHL1+/+ mice. The underlying mechanisms probably involve better axonal regeneration and sprouting in the lumbar spinal cord after injury, due to the absence of unexpected adverse CHL1-CHL1 homophilic interactions in the mutant [8]. In addition, motoneuronal excitability after injury, as measured by H-reflex analysis, was higher in female CHL1−/− than in female CHL1+/+ mice, possibly due to fewer inhibitory interneurons in CHL1−/− mice [39]. We tried to stain spinal cords with two other interneuron markers (GAD67, calbindin), but the staining quality was not good enough for reliable quantifications. We emphasize that all our conclusions regarding functional recovery after injury apply to female, but not to male CHL1−/− mice. However, as there was no correlation between the functional recovery and the number of parvalbumin-expressing interneurons, we cannot predict if the loss of these interneurons would also occur in CHL1−/− male mice. Since the mechanisms underlying the selective loss of parvalbumin-expressing neurons in CHL1−/− mice remain elusive, we propose that different interaction partners of CHL1, such as integrins, could play a role [42].

Further experiments are needed to establish the exact timeline of degeneration upon injury. The time-point of 6 weeks post-injury in our study was chosen as there was no further recovery of motor function after this time in CHL1+/+ or CHL1−/− mice [8]. This, however, does not preclude further progressive degeneration after this time, which probably depends on the severity of the injury. Another interesting point to be addressed in further studies would be to determine at which time after injury the CHL1 mutation impairs parvalbumin-expressing interneurons. 

This work was part of the doctoral thesis [43].

## 5. Conclusions

Both neuronal and glial cell populations in the lumbar spinal cord, as well as the spinal cord volume, are affected 6 weeks after low-thoracic spinal cord injury. The spinal cord volume and the numbers of motoneurons, parvalbumin-expressing interneurons, and oligodendrocytes in the white matter were reduced upon injury, whereas the numbers of microglia/macrophages were increased in the gray matter, in both genotypes. After injury, changes in lumbar spinal cord volume and in neuronal and glial populations of CHL1+/+ mice were similar to those of CHL1−/− mice, with the exception of parvalbumin-positive interneurons, which are lost to a higher degree in the absence of CHL1. It will be interesting to examine cell death and proliferation in the lumbar spinal cord at different times after injury. This could be helpful for designing future therapies, such as treatment with neuroprotective agents for motoneurons and interneurons. However, cell replacement as a therapeutic approach to ameliorate degeneration will not be amenable in the near future. Yet, application of L1 agonistic small organic compounds that pass the blood–brain barrier and are FDA approved, appears to be a realistic approach to ameliorate not only injury-related abnormalities, but also different types of neurodegeneration. 

## Figures and Tables

**Figure 1 biomolecules-15-01247-f001:**
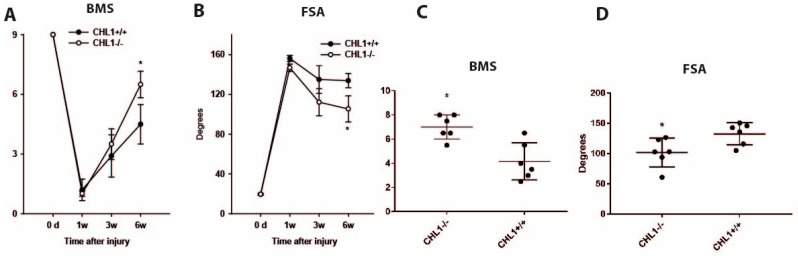
Recovery of locomotor function as estimated by Basso Mouse Score (**A**,**C**) and foot stepping angle (**B**,**D**) over time (**A**,**B**) and at 6 weeks after injury (**C**,**D**). (**A**,**B**) Shown are mean values ± standard error of the mean for BMS (**A**) and FSA (**B**) before injury (0 day) and at 1 week, and 3 and 6 weeks after injury. Asterisks represent the difference between genotypes (*p* < 0.05; Two-way ANOVA for repeated measures with Holm–Sidak post hoc test, n = 6 mice/group). (**C**,**D**) Scatter dot plots show mean ± SD values for BMS (**C**) and FSA (**D**) at 6 weeks after injury. Asterisks (*) represent the difference between genotypes (*p* < 0.05; Two-way ANOVA for repeated measures with Holm–Sidak post hoc test, n = 6 mice/group).

**Figure 2 biomolecules-15-01247-f002:**
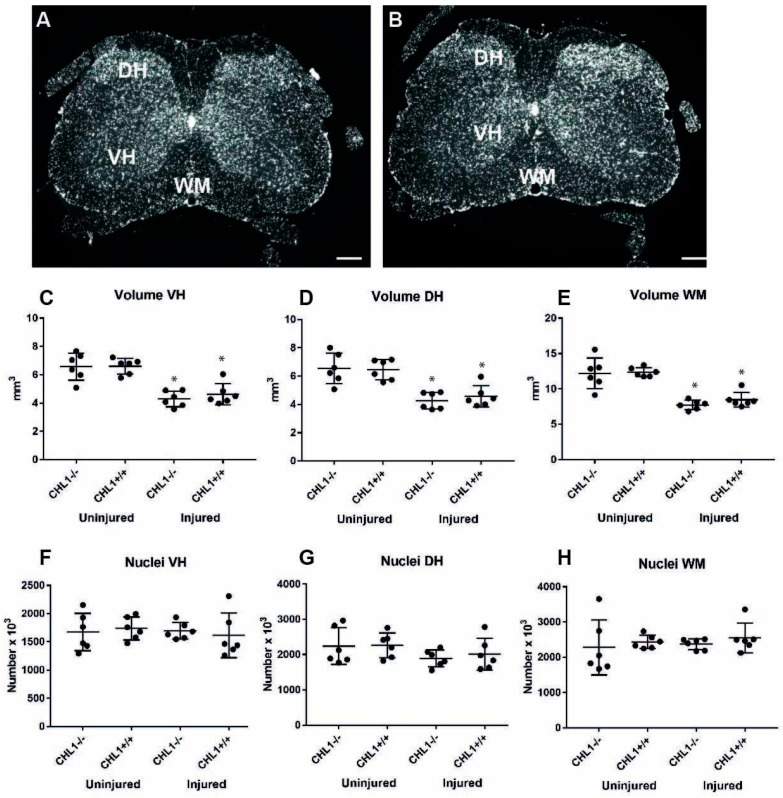
Volume and total number of cell nuclei without (**A**) and after (**B**) injury. Representative images of DAPI nuclear staining of uninjured (**A**) and injured (**B**) CHL1+/+ spinal cords. Red lines delineate ventral horns (VH), dorsal horns (DH), and the white matter (WM). Scale bars: 200 µm. (**C**–**E**) Scatter dot plots show mean ± SD values for volumes of the lumbar spinal cord ventral horn (**C**), dorsal horn (**D**), and the white matter (**E**). Asterisks represent the difference between uninjured and injured, and hashtags represent the difference between genotypes (*p* < 0.05; one-way ANOVA with Holm–Sidak post hoc test, n = 6 mice / group). (**F**–**H**) Scatter dot plots show mean ± SD values for the numbers of DAPI+ nuclei in the lumbar spinal cord ventral horn (**F**), dorsal horn (**G**), and white matter (**H**). Asterisks (*) indicate the difference between uninjured and injured animals, and hashtags represent the difference between genotypes (*p* < 0.05; one-way ANOVA with Holm–Sidak post hoc test, n = 6 mice/group).

**Figure 3 biomolecules-15-01247-f003:**
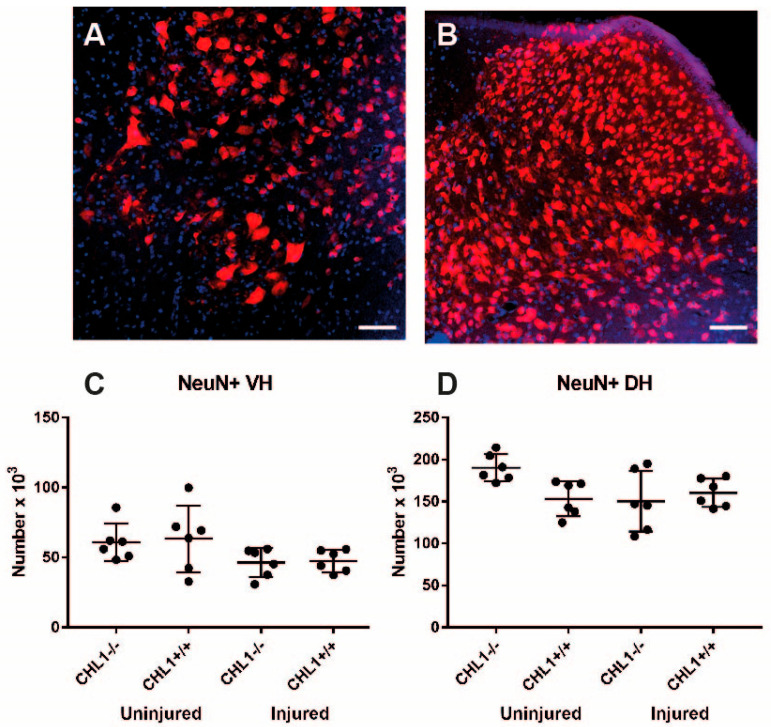
NeuN-positive neurons without and after injury. Representative images of the NeuN immunostaining (red) of the uninjured CHL1+/+ lumbar spinal cord ventral horn (**A**) and dorsal horn (**B**) neurons. DAPI (blue) indicates nuclei. Scale bars: 25 µm. (**C**,**D**) Scatter dot plots show mean ± SD values for the number of NeuN-positive cells in the ventral horn (**C**) and dorsal horn (**D**). There were no significant differences between groups (*p* > 0.05; one-way ANOVA with Holm–Sidak post hoc test, n = 6 mice/group).

**Figure 4 biomolecules-15-01247-f004:**
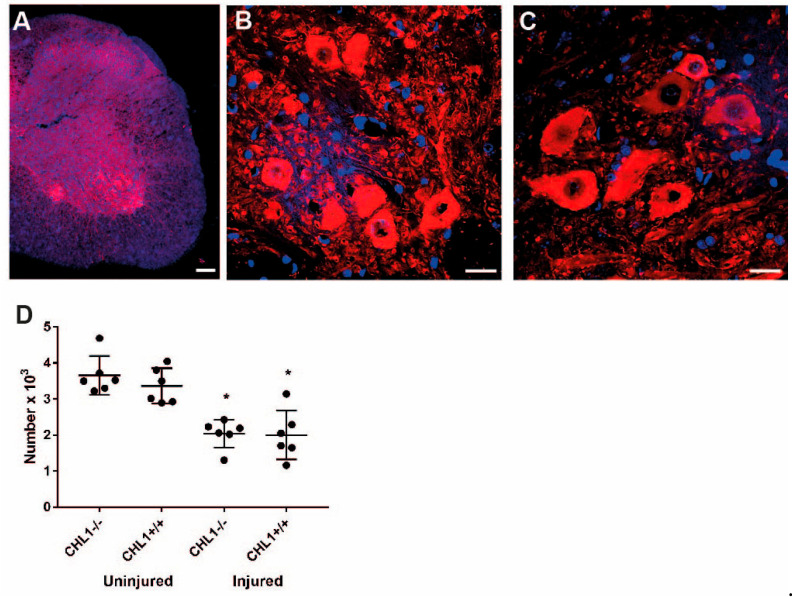
ChAT+ motoneurons without and after injury. Representative images of ChAT immunostaining (red) of uninjured (**A**,**B**) and injured (**C**) CHL1+/+ ventral horns. Nuclei are stained with DAPI (blue). Scale bars: 200 µm (**A**), 25 µm (**B**,**C**). (**D**) Scatter dot plots show mean ± SD for the number of ChAT+ motoneurons in the lumbar spinal cord ventral horn. Asterisks (*) indicate the difference between uninjured and injured (*p* < 0.05; one-way ANOVA with Holm–Sidak post hoc test, n = 6 mice/group).

**Figure 5 biomolecules-15-01247-f005:**
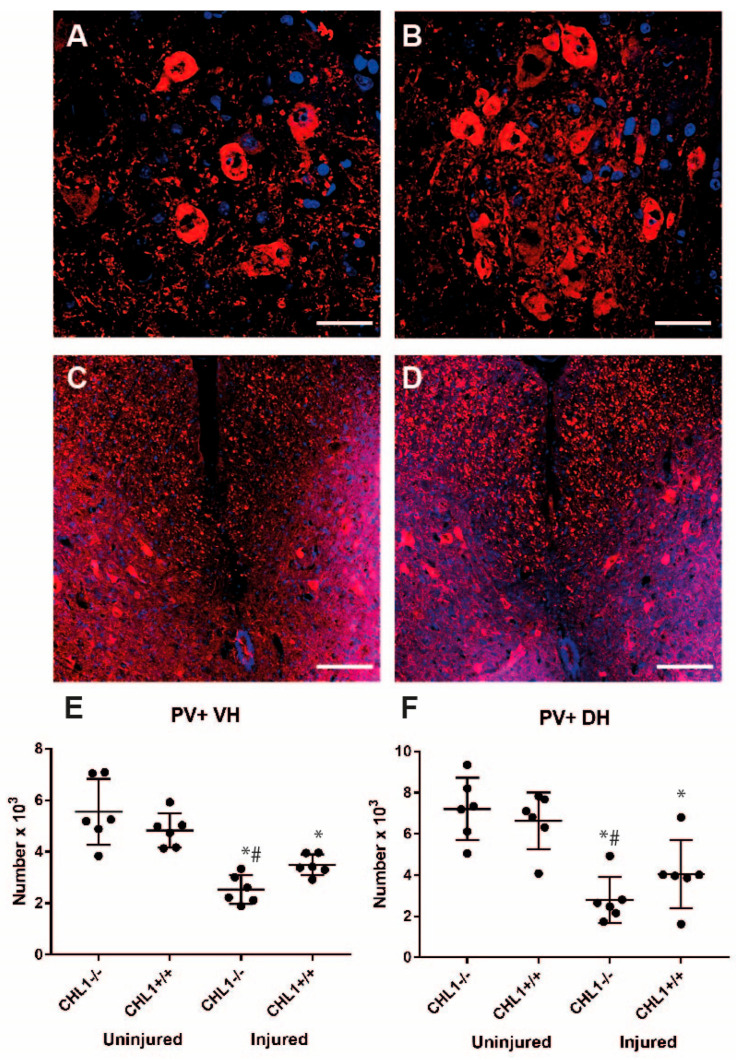
Parvalbumin (PV)-positive interneurons without and after injury. (**A**–**D**) Representative images of PV immunostaining (red) of uninjured CHL1−/− (**A**,**C**) and CHL1+/+ (**B**,**D**), lumbar spinal cord ventral horn (**A**,**B**), and dorsal horn (**C**,**D**) interneurons. Nuclei are stained with DAPI (blue). Scale bars: 25 µm. (**E**,**F**) Scatter dot plots show mean ± SD for the number of PV+ cells in the lumbar spinal cord ventral horn (**E**) and dorsal horn (**F**). Asterisks indicate the difference between uninjured and injured, and hashtags represent the difference between genotypes (*p* < 0.05; one-way ANOVA with Holm–Sidak post hoc test, n = 6 mice/group).

**Figure 6 biomolecules-15-01247-f006:**
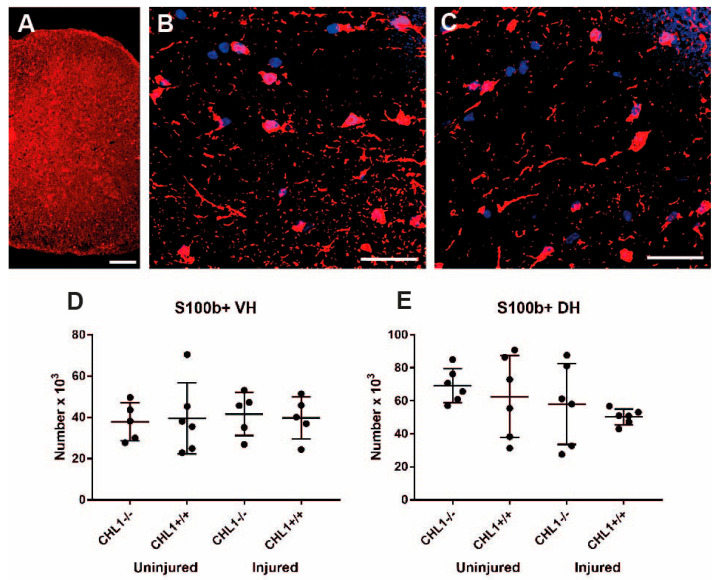
S100b+ astrocytes without and after injury. Representative images of the S100b immunostaining (red) of the uninjured CHL1+/+ lumbar spinal cord hemisection (**A**), ventral horn (**B**), and dorsal horn (**C**) astrocytes. Nuclei are stained with DAPI (blue). Scale bars: 25 µm. (**D**,**E**) Scatter dot plots show mean ± SD for the number of S100b+ astrocytes in the lumbar spinal cord ventral horn (**D**) and dorsal horn (**E**). There were no significant differences between groups (*p* > 0.05; one-way ANOVA with Holm–Sidak post hoc test, n = 6 mice/group).

**Figure 7 biomolecules-15-01247-f007:**
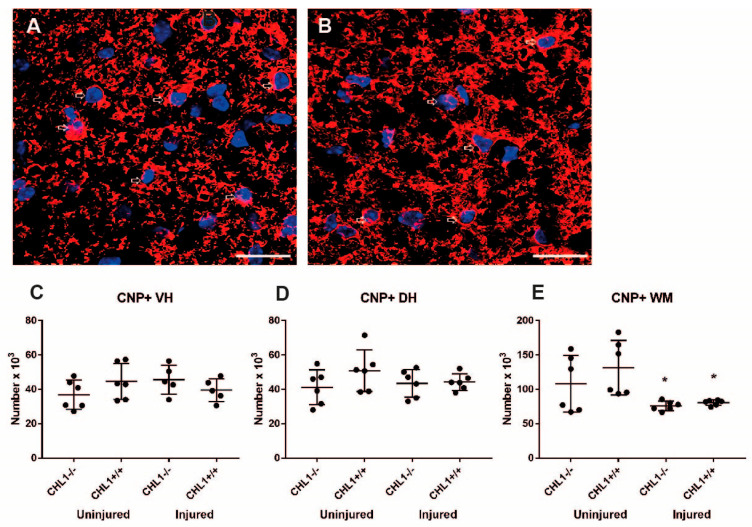
CNP+ oligodendrocytes without and after injury. Representative images of the CNP immunostaining (red) of the uninjured (**A**) and injured (**B**) CHL1+/+ white matter oligodendrocytes. Nuclear staining is indicated in blue. Arrows point to immunolabeled cells. Scale bars: 20 µm. (**C**–**E**) Scatter dot plots show mean ± SD for the number of CNP+ cells in the lumbar spinal cord ventral horn (**C**), dorsal horn (**D**), and white matter (**E**). Asterisks (*) indicate the difference between uninjured and injured groups (*p* < 0.05; one-way ANOVA with Holm–Sidak post hoc test, n = 6 mice/group).

**Figure 8 biomolecules-15-01247-f008:**
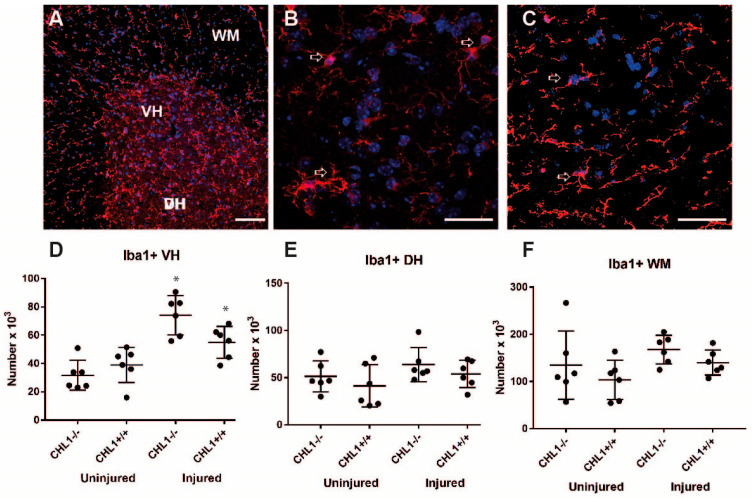
Iba1-positive microglia/macrophages in the lumbar spinal cord after injury (**A**–**C**). Representative images of the Iba1 immunostaining (red) of the injured CHL1+/+ lumbar spinal cord hemisection (**A**), gray matter (**B**), and white matter (**C**). Nuclei are shown in blue. Arrows point to immunolabeled cells. Scale bars: 50 µm (**A**), 25 µm (**B**,**C**). (**C**,**D**) Scatter dot plots show mean ± SD for the number of Iba1+ microglia/macrophages in the lumbar spinal cord ventral horn (**D**), dorsal horn (**E**) and the white matter (**F**). Asterisks (*) indicate the difference between uninjured and injured (*p* < 0.05; one-way ANOVA with Holm–Sidak post hoc test, n = 6 mice/group).

**Table 1 biomolecules-15-01247-t001:** Scheme showing the numbers of neurons and glial cells in the lumbar spinal cord after low-thoracic cord injury.

Change	Injured vs. Uninjured CHL1+/+	Injured CHL1−/− vs. Injured CHL1+/+
Volume	↓	=
All cells/DAPI	=	=
All neurons/NeuN	=	=
Motoneurons/ChAT	↓	=
Interneurons/PV	↓	↓
Astrocytes/S100b	=	=
Oligodendrocytes/CNP	WM ↓ GM =	=
Microglia/Iba ^1^	VH ↑ DH = WM =	=

^1^ Legend: WM—white matter; GM—gray matter; VH—ventral horn; DH—dorsal horn. ↓—decrease; ↑—increase; =—no difference.

## Data Availability

Data is contained within the article or Appendix A.

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
