# Peer review of "Consequences of Adhesion Molecule Close Homolog of L1 Deficiency for Neurons and Glial Cells in the Mouse Spinal Cord After Injury"

_biomolecules, 2025, doi:10.3390/biom15091247_

Round 1

Reviewer 1 Report

Comments and Suggestions for Authors

In this work, Jakovcevski and colleagues investigate how the deficiency of the cell adhesion molecule Close Homolog of L1 (CHL1) affects neural cell populations in a mouse spinal cord injury (SCI) model. Using a low-thoracic compression injury in adult female mice, they quantified major neuronal and glial cell types in the lumbar spinal cord 6 weeks post-injury. Six weeks after SCI, CHL1 knock-out (CHL1−/−) and wild-type littermates exhibited a loss of lumbar gray and white matter volume, accompanied by reduced total motoneuron and PV+ interneuron numbers compared to uninjured controls, with CHL1-deficient mice suffering a greater loss of interneurons than wild-type mice. The authors conclude that CHL1 deficiency exacerbates certain injury-induced pathologies, such as interneuron loss and oligodendrocyte reduction, which are not seen in wild-type mice.

Past research had produced conflicting results regarding CHL1’s role in recovery. For example, this group reported in 2007 that CHL1 is upregulated in the glial scar and can limit axonal regrowth, suggesting CHL1 removal might improve recovery. However, more recent work by Theis et al. (2022) found that in male mice, CHL1 deficiency made no difference in functional recovery. The use of an unbiased stereological counting method here is a notable advance, as it resolves controversies from older studies that relied on potentially biased cell counts.

When placed in the context of recent literature, the contribution appears incremental but valuable. It confirms that even in chronic stages post-SCI, there is measurable loss of specific neurons and glia far from the injury, and it implicates CHL1 in modulating these changes. This aligns with the growing recognition that interneurons and network plasticity below the lesion are critical to recovery. The CHL1 adhesion molecule itself has been a focus of regeneration research: for instance, overexpression of the related L1 CAM in injured spinal cord promotes axonal regrowth, remyelination, and functional recovery, and administration of small molecule L1-mimetics has improved recovery in preclinical SCI models. Given this background, the present study’s indication that CHL1 supports the survival of certain neurons is biologically plausible, as it aligns with the concept that adhesion molecules can create a more permissive environment for regeneration. The findings suggest that, contrary to some expectations, CHL1’s presence may be protective for interneurons and oligodendrocytes after injury, at least in female mice, highlighting the complexity of targeting adhesion molecules for therapy.

The study design is generally sound. Focusing on female mice is justified by practical considerations (easier postoperative care; male mice often have urinary complications post-SCI) and by prior evidence that CHL1 effects were pronounced in females. Nonetheless, this choice does limit the scope (see “Major Comments” below).

A major strength of this paper is the use of design-based stereology to quantify cells. The authors implemented an optical dissector/Cavalieri method with systematic random sampling of every 250 µm section through the lumbar enlargement. One minor detail that could be added is an estimate of counting precision (e.g., Gundersen’s coefficient of error), but overall the stereological approach appears properly implemented and is a highlight of the methodological rigor.

The images in the figures of the immunohistochemical experiments are of very poor quality, making proper assessment difficult.

Some of the authors’ conclusions border on over-interpretation. They assert that “CHL1 deficiency causes deficits in spinal cord regeneration”. Strictly speaking, their study demonstrates structural deficits (greater cell loss) in the CHL1-deficient spinal cord, but “regeneration” implies axonal regrowth or functional recovery, which was not directly assessed. The improved behavioral recovery seen in CHL1-knockout mice by others complicates this narrative. The authors acknowledge the paradox that CHL1−/− mice can recover as well as or better than wild-types despite what would seem like a disadvantage (loss of interneurons). Their speculative resolution – that CHL1 deficiency both removes a regeneration brake and increases circuit excitability – is intriguing and logically consistent, but remains unproven by the data in this manuscript. For instance, no tracers or histological evidence of enhanced axonal sprouting in the lumbar cord were provided, and no functional tests (beyond literature references) were performed to correlate with the observed histology. Therefore, while the speculative discussion is intellectually engaging and not implausible, it somewhat oversteps what the results alone can support. A more cautious phrasing would be that CHL1 deficiency “exacerbates distal neuron loss and oligodendrocyte loss,” and that this might in part be offset functionally by other adaptive mechanisms (rather than framing it as definitive “deficits in regeneration”).

The authors’ interpretations about cell-type vulnerability (e.g. CHL1-deficient interneurons being more vulnerable, analogous to observations in aging CHL1 mutant brains) are well-argued. Their speculation linking those findings to functional outcome is clearly labeled as such (they use terms like “could be explained by…” and “underlying mechanisms probably involve…”), which is appropriate in a discussion section. The balance between results and speculation is generally acceptable, but some claims (particularly the use of the term “regeneration”) could be toned down or backed by additional data. The manuscript would benefit from a brief acknowledgment that functional regeneration was not measured here, and thus any implications for recovery are indirect. Overall, the conclusions drawn are reasonable given the data, and the authors do a commendable job reconciling their results with the mixed literature. Their forward-looking suggestion that L1-mimetic compounds (some of which are blood–brain barrier permeable and FDA-approved) might be tested to promote recovery is forward-thinking and connects the study to potential therapeutic avenues, albeit without experimental evidence in this paper.

There are only a few minor language issues to note: for example, a typographical error in the Results where “uninjured” is misspelled as “unjunured”, and, in a sentence in the Introduction the phrasing is a bit unclear: “leading to better recovery after injury in female, but not in male CHL1-deficient (CHL1−/−) mice [8,9]”. For clarity, it could be rewritten as “leading to better post-injury recovery in female CHL1−/− mice, but not in males.”

In one figure legend, an asterisk was indicated but the text read “p > 0.05”, which appears to be an editorial mistake (probably it should be p < 0.05).

Major Comments

  • A significant limitation of this study is that only female mice were used. The authors did this for justifiable practical reasons, but it nonetheless means the findings may not extend to male physiology. This is important because CHL1’s impact on recovery has been shown to differ by sex. The authors should explicitly acknowledge in the Discussion that conclusions apply to females, and discuss whether similar cell loss patterns would be expected in males. This is especially relevant since male CHL1-knockouts did not show improved recovery in prior studies – perhaps male mice would also not show the heightened interneuron loss observed here? Addressing this point would strengthen the manuscript’s conclusions and transparency about its scope.
  • The study focuses on morphological outcomes (cell counts and volumes) but does not directly assess functional recovery or axon regeneration. While behavioral monitoring was mentioned, no locomotor score results are reported, and there were no tracers or immunostains (e.g., for serotonergic fibers or corticospinal axons) to evaluate axonal growth past the injury. Given that the paper concludes CHL1 deficiency causes “deficits in regeneration,” it is a major oversight not to measure any markers of axon regrowth or functional connectivity. The authors are encouraged to temper claims about “regeneration” or, if possible, provide some supplementary data or references demonstrating whether CHL1−/− mice have differences in axon sprouting in the lumbar cord. At minimum, the discussion should clarify that “regeneration” was inferred from cell-survival outcomes, not directly observed. This distinction is important to avoid over-interpretation.
  • The use of one-way ANOVA with post-hoc tests is acceptable, but the manuscript would benefit from a more direct statistical analysis of the interaction between injury and genotype. For example, a two-way ANOVA could confirm whether the genotype effect on interneuron number is statistically significant only in the injured condition (i.e., a significant interaction). As presented, the data imply this, but the authors do not report an interaction p-value. If not re-analysis, the authors should at least explicitly state in text that “no significant genotype effect was present in uninjured mice, whereas a significant effect emerged after injury for PV+ interneurons”. This would clarify the interpretation. Similarly, it should be made clear that for most cell types (motoneurons, astrocytes, etc.), CHL1 genotype had no effect (the knockout parallels wild-type). This point is somewhat lost in the text and figures, as the emphasis is on the difference. Strengthening the description of negative results (e.g., “there was no difference in astrocyte or microglial numbers between genotypes, indicating CHL1 does not affect glial proliferation after injury”) would provide a more balanced view.
  • There is an underlying contradiction that warrants further discussion: CHL1-deficient mice in this experiment exhibit more tissue damage (greater interneuron and oligodendrocyte loss), yet previous studies have shown equal or better behavioral outcomes in such mice. The authors do mention a hypothesis that interneuron loss might heighten reflexes and thus mask functional deficits, but this is a critical point that deserves more emphasis. As a major comment, I suggest the authors expand on how their structural findings align (or not) with functional recovery data. Could the loss of PV+ interneurons (likely inhibitory interneurons) lead to spasticity or disinhibition that improves certain locomotor scores at the cost of refined motor control? Is it possible that CHL1-/- mice have compensatory mechanisms (e.g., reorganization of circuits) that overcome the greater cell loss? By exploring these questions, the manuscript will provide a more comprehensive understanding. If the authors have any data on locomotor recovery from these same mice (even if showing no genotype difference), reporting it would be valuable to the reader.
  • The study examines a single chronic time point (6 weeks post-injury). A major point for future investigation is when these differences in interneuron number arise. The authors themselves suggest examining multiple time points going forward. As a reviewer, I emphasize this suggestion: including an earlier post-injury time (e.g., 1–2 weeks) could help distinguish whether CHL1 deficiency accelerates early cell death versus causing a failure of later cell replacement. Although new experiments are not required for the current manuscript, the authors should discuss this in more detail. For instance, if interneuron loss in CHL1-/- mice is an early event, interventions would need to target acute neuroprotection; if it’s a chronic degeneration, other strategies apply. Elaborating on this in the Discussion would enhance the impact and relevance of the findings, showing the authors have considered the translational timing aspect.

Author Response

In this work, Jakovcevski and colleagues investigate how the deficiency of the cell adhesion molecule Close Homolog of L1 (CHL1) affects neural cell populations in a mouse spinal cord injury (SCI) model. Using a low-thoracic compression injury in adult female mice, they quantified major neuronal and glial cell types in the lumbar spinal cord 6 weeks post-injury. Six weeks after SCI, CHL1 knock-out (CHL1−/−) and wild-type littermates exhibited a loss of lumbar gray and white matter volume, accompanied by reduced total motoneuron and PV+ interneuron numbers compared to uninjured controls, with CHL1-deficient mice suffering a greater loss of interneurons than wild-type mice. The authors conclude that CHL1 deficiency exacerbates certain injury-induced pathologies, such as interneuron loss and oligodendrocyte reduction, which are not seen in wild-type mice.

Past research had produced conflicting results regarding CHL1’s role in recovery. For example, this group reported in 2007 that CHL1 is upregulated in the glial scar and can limit axonal regrowth, suggesting CHL1 removal might improve recovery. However, more recent work by Theis et al. (2022) found that in male mice, CHL1 deficiency made no difference in functional recovery. The use of an unbiased stereological counting method here is a notable advance, as it resolves controversies from older studies that relied on potentially biased cell counts.

R: We would like to point out that there is a difference between genders for CHL1 knockout mice in their response to spinal cord injury: females recover better, whereas males showed no difference to the wild-type. The only difference in males, but not in females, were the numbers of neutrophils (Jakovcevski et al., 2007, ref. 8; Theis et al, 2022, ref. 9).

When placed in the context of recent literature, the contribution appears incremental but valuable. It confirms that even in chronic stages post-SCI, there is measurable loss of specific neurons and glia far from the injury, and it implicates CHL1 in modulating these changes. This aligns with the growing recognition that interneurons and network plasticity below the lesion are critical to recovery. The CHL1 adhesion molecule itself has been a focus of regeneration research: for instance, overexpression of the related L1 CAM in injured spinal cord promotes axonal regrowth, remyelination, and functional recovery, and administration of small molecule L1-mimetics has improved recovery in preclinical SCI models. Given this background, the present study’s indication that CHL1 supports the survival of certain neurons is biologically plausible, as it aligns with the concept that adhesion molecules can create a more permissive environment for regeneration. The findings suggest that, contrary to some expectations, CHL1’s presence may be protective for interneurons and oligodendrocytes after injury, at least in female mice, highlighting the complexity of targeting adhesion molecules for therapy.

The study design is generally sound. Focusing on female mice is justified by practical considerations (easier postoperative care; male mice often have urinary complications post-SCI) and by prior evidence that CHL1 effects were pronounced in females. Nonetheless, this choice does limit the scope (see “Major Comments” below).

A major strength of this paper is the use of design-based stereology to quantify cells. The authors implemented an optical dissector/Cavalieri method with systematic random sampling of every 250 µm section through the lumbar enlargement. One minor detail that could but overall the stereological approach appears properly implemented and is a highlight of the methodological rigor.

R: We thank the Reviewer for the appreciation of our methodology. We have used the same sampling method in some of our publications (Jakovcevski et al., 2009; Mehanna et al., 2010; Wu et al., 2012; Schmalbach et al., 2015).

The images in the figures of the immunohistochemical experiments are of very poor quality, making proper assessment difficult.

R: We have now made all possible efforts to present better images.

Some of the authors’ conclusions border on over-interpretation. They assert that “CHL1 deficiency causes deficits in spinal cord regeneration”. Strictly speaking, their study demonstrates structural deficits (greater cell loss) in the CHL1-deficient spinal cord, but “regeneration” implies axonal regrowth or functional recovery, which was not directly assessed. The improved behavioral recovery seen in CHL1-knockout mice by others complicates this narrative. The authors acknowledge the paradox that CHL1−/− mice can recover as well as or better than wild-types despite what would seem like a disadvantage (loss of interneurons). Their speculative resolution – that CHL1 deficiency both removes a regeneration brake and increases circuit excitability – is intriguing and logically consistent, but remains unproven by the data in this manuscript. For instance, no tracers or histological evidence of enhanced axonal sprouting in the lumbar cord were provided, and no functional tests (beyond literature references) were performed to correlate with the observed histology. Therefore, while the speculative discussion is intellectually engaging and not implausible, it somewhat oversteps what the results alone can support. A more cautious phrasing would be that CHL1 deficiency “exacerbates distal neuron loss and oligodendrocyte loss,” and that this might in part be offset functionally by other adaptive mechanisms (rather than framing it as definitive “deficits in regeneration”).

R: We now added the results of functional recovery to the manuscript (new Figure 1). As we used transversal tissue sections, it would be very difficult to assess axonal regeneration. Nevertheless, we expect that it is similar to what we reported on female mice (Jakovcevski et al., 2007). We have now rephrased the text by using the reviewer’s suggestion. (lines 43-44; 254-255)

The authors’ interpretations about cell-type vulnerability (e.g. CHL1-deficient interneurons being more vulnerable, analogous to observations in aging CHL1 mutant brains) are well-argued. Their speculation linking those findings to functional outcome is clearly labeled as such (they use terms like “could be explained by…” and “underlying mechanisms probably involve…”), which is appropriate in a discussion section. The balance between results and speculation is generally acceptable, but some claims (particularly the use of the term “regeneration”) could be toned down or backed by additional data. The manuscript would benefit from a brief acknowledgment that functional regeneration was not measured here, and thus any implications for recovery are indirect. Overall, the conclusions drawn are reasonable given the data, and the authors do a commendable job reconciling their results with the mixed literature. Their forward-looking suggestion that L1-mimetic compounds (some of which are blood–brain barrier permeable and FDA-approved) might be tested to promote recovery is forward-thinking and connects the study to potential therapeutic avenues, albeit without experimental evidence in this paper.

R: Some statements have been tuned down.

There are only a few minor language issues to note: for example, a typographical error in the Results where “uninjured” is misspelled as “unjunured”, and, in a sentence in the Introduction the phrasing is a bit unclear: “leading to better recovery after injury in female, but not in male CHL1-deficient (CHL1−/−) mice [8,9]”. For clarity, it could be rewritten as “leading to better post-injury recovery in female CHL1−/− mice, but not in males.”

R: We apologize for the language issues, which we have now corrected. We also rewrote the sentence as suggested. (lines 70-71)

In one figure legend, an asterisk was indicated but the text read “p > 0.05”, which appears to be an editorial mistake (probably it should be p < 0.05).

R: We apologize for this error in the legend of Figure 3, which we have now corrected.

Major Comments

  • A significant limitation of this study is that only female mice were used. The authors did this for justifiable practical reasons, but it nonetheless means the findings may not extend to male physiology. This is important because CHL1’s impact on recovery has been shown to differ by sex. The authors should explicitly acknowledge in the Discussion that conclusions apply to females, and discuss whether similar cell loss patterns would be expected in males. This is especially relevant since male CHL1-knockouts did not show improved recovery in prior studies – perhaps male mice would also not show the heightened interneuron loss observed here? Addressing this point would strengthen the manuscript’s conclusions and transparency about its scope.

R: We have now added the following sentence to the Discussion: “We emphasize that our conclusions regarding functional recovery after injury apply to female, but not to male CHL1-/- mice. However, since there was no correlation between functional recovery and number of parvalbumin-expressing interneurons, we cannot predict if the loss of these interneurons would occur also in CHL1-/- male mice.” (lines 515-519)

  • The study focuses on morphological outcomes (cell counts and volumes) but does not directly assess functional recovery or axon regeneration. While behavioral monitoring was mentioned, no locomotor score results are reported, and there were no tracers or immunostains (e.g., for serotonergic fibers or corticospinal axons) to evaluate axonal growth past the injury. Given that the paper concludes CHL1 deficiency causes “deficits in regeneration,” it is a major oversight not to measure any markers of axon regrowth or functional connectivity. The authors are encouraged to temper claims about “regeneration” or, if possible, provide some supplementary data or references demonstrating whether CHL1−/− mice have differences in axon sprouting in the lumbar cord. At minimum, the discussion should clarify that “regeneration” was inferred from cell-survival outcomes, not directly observed. This distinction is important to avoid over-interpretation.

R: We have now added the results for locomotor recovery of mice that were used for morphological analyses. In the abstract, we changed “regeneration” to “structural outcome”. Additionally, we rephrased “regeneration” to “recovery” where necessary.

  • The use of one-way ANOVA with post-hoc tests is acceptable, but the manuscript would benefit from a more direct statistical analysis of the interaction between injury and genotype. For example, a two-way ANOVA could confirm whether the genotype effect on interneuron number is statistically significant only in the injured condition (i.e., a significant interaction). As presented, the data imply this, but the authors do not report an interaction p-value. If not re-analysis, the authors should at least explicitly state in text that “no significant genotype effect was present in uninjured mice, whereas a significant effect emerged after injury for PV+ interneurons”. This would clarify the interpretation. Similarly, it should be made clear that for most cell types (motoneurons, astrocytes, etc.), CHL1 genotype had no effect (the knockout parallels wild-type). This point is somewhat lost in the text and figures, as the emphasis is on the difference. Strengthening the description of negative results (e.g., “there was no difference in astrocyte or microglial numbers between genotypes, indicating CHL1 does not affect glial proliferation after injury”) would provide a more balanced view.

R: We have re-analyzed the data for PV+ interneurons with two-way ANOVA and report on significant interaction between factors “genotype” and “injury” (lines 292-296). Moreover, we report on the p value. Furthermore, we emphasized the lack of the effect of genotype on other cell types. (lines 496-498)

  • There is an underlying contradiction that warrants further discussion: CHL1-deficient mice in this experiment exhibit more tissue damage (greater interneuron and oligodendrocyte loss), yet previous studies have shown equal or better behavioral outcomes in such mice. The authors do mention a hypothesis that interneuron loss might heighten reflexes and thus mask functional deficits, but this is a critical point that deserves more emphasis. As a major comment, I suggest the authors expand on how their structural findings align (or not) with functional recovery data. Could the loss of PV+ interneurons (likely inhibitory interneurons) lead to spasticity or disinhibition that improves certain locomotor scores at the cost of refined motor control? Is it possible that CHL1-/- mice have compensatory mechanisms (e.g., reorganization of circuits) that overcome the greater cell loss? By exploring these questions, the manuscript will provide a more comprehensive understanding. If the authors have any data on locomotor recovery from these same mice (even if showing no genotype difference), reporting it would be valuable to the reader.

R: We have now added the following sentences to the Discussion: “However, reduced inhibition from parvalbumin-expressing interneurons could counterintuitively increase locomotor function due to a higher spasticity in the lower limbs or cause further, still unknown compensatory remodeling mechanisms in the spinal cord circuitry, which could lead to better locomotion. It is noteworthy that CHL1-/- female mice show better preservation or better regeneration of cholinergic synaptic terminals around motoneurons [8].” (lines 444-450)

  • The study examines a single chronic time point (6 weeks post-injury). A major point for future investigation is when these differences in interneuron number arise. The authors themselves suggest examining multiple time points going forward. As a reviewer, I emphasize this suggestion: including an earlier post-injury time (e.g., 1–2 weeks) could help distinguish whether CHL1 deficiency accelerates early cell death versus causing a failure of later cell replacement. Although new experiments are not required for the current manuscript, the authors should discuss this in more detail. For instance, if interneuron loss in CHL1-/- mice is an early event, interventions would need to target acute neuroprotection; if it’s a chronic degeneration, other strategies apply. Elaborating on this in the Discussion would enhance the impact and relevance of the findings, showing the authors have considered the translational timing aspect.

R:  We have now added the following sentence to the Discussion: “Further experiments are needed to establish the exact timeline of neurodegeneration upon injury. The time-point of 6 weeks post-injury in our study was chosen as there was no further recovery of motor function after this time in CHL1+/+ or CHL1-/- mice [8]. This, however, does not preclude further progressive degeneration after this time, which probably depends on the severity of injury. Another interesting point to be addressed in further study would be to determine at which time after injury the CHL1 mutation impairs parvalbumin-expressing interneurons.” (lines 525-532)

Reviewer 2 Report

Comments and Suggestions for Authors

This study provides a detailed stereological analysis of neuronal and glial cell populations in the lumbar spinal cord following low-thoracic injury, using CHL1-deficient and wild-type mice. While the methodology is sound and the model is well controlled, the experimental design is relatively straightforward. The authors do not provide fundamentally new mechanistic insights. Several major concerns need to be addressed before the manuscript can be considered for publication.

Major points:

Why did the authors choose to analyze the spinal cord specifically at 6 weeks post-injury? Did they perform any time-course studies (e.g., at 1 week, 3 weeks, or later time points) to assess the dynamics of these changes?

How do the authors explain the selective vulnerability of parvalbumin-positive interneurons in CHL1-deficient mice?

It would be important to examine markers of cell death (e.g., apoptosis) or increased microglial phagocytic activity around these interneurons.

While the study focused on parvalbumin-positive interneurons, it would also be important to analyze other interneuron subtypes (e.g., calbindin-positive or GABAergic populations).

Regarding oligodendrocytes, did the authors assess myelin integrity using myelin-specific stains (Luxol Fast Blue or MBP ihc)?

Did the authors assess functional outcomes, such as locomotor performance (e.g., Basso Mouse Scale or other behavioral tests, f.e. video-based kinematic analysis)?

Minor points:

“Among the cell adhesion molecules of interest for regeneration is the close homolog 66 of adhesion molecule L1 (CHL1), which was shown to be upregulated by astrocytes at the 67 lesion site after spinal cord injury [8]. This unexpected increase was considered to hamper 68 regeneration through homophilic CHL1-CHL1 interactions between axons and astrocytes, 69 leading to better recovery after injury in female, but not in male CHL1-deficient (CHL1-/-70 ) mice [8,9].” This section is somewhat difficult to follow due to its complex sentence structure and dense information. Rephrasing it for clarity and simplicity would greatly improve readability and help convey the intended message more effectively.

The quality of microphotographs of immunodetection, particularly those of high power, needs substantial improvement. Please add better images to the manuscript and also quantify your staining as it is state-of-the-art.

Author Response

This study provides a detailed stereological analysis of neuronal and glial cell populations in the lumbar spinal cord following low-thoracic injury, using CHL1-deficient and wild-type mice. While the methodology is sound and the model is well controlled, the experimental design is relatively straightforward. The authors do not provide fundamentally new mechanistic insights. Several major concerns need to be addressed before the manuscript can be considered for publication.

R: We are very grateful to the Reviewer for the appreciation of our manuscript.

Major points:

Why did the authors choose to analyze the spinal cord specifically at 6 weeks post-injury? Did they perform any time-course studies (e.g., at 1 week, 3 weeks, or at later time points) to assess the dynamics of these changes?

R: Our previous work on spinal cord injury with the same mouse strains has shown that recovery of motor function progresses until approximately 6 weeks after injury. We therefore analyzed locomotor recovery, and observed that after 6 weeks recovery had reached a plateau level in both genotypes. With respect to this observation, we consider our time point as “chronic” injury. For the time points before (1 and 3 weeks post-injury), we followed locomotor recovery, but as no difference in recovery was seen, we assumed that the difference would not be represented by a structural outcome. We also discuss this now: “Further experiments are needed to establish the exact timeline of neurodegeneration upon injury. The time-point of 6 weeks post-injury in our study was chosen as there was no further recovery of motor function after this time in CHL1+/+ or CHL1-/- mice [8]. This, however, does not preclude further progressive degeneration after this time, which probably depends on the severity of injury. Another interesting point to be addressed in further study would be to determine at which time after injury the CHL1 mutation impairs parvalbumin-expressing interneurons. (lines 525-529)

How do the authors explain the selective vulnerability of parvalbumin-positive interneurons in CHL1-deficient mice?

R: We now speculated further on the selective vulnerability of PV+ interneurons in CHL1-/- mice. We added the following sentence to the Discussion: “Since the mechanisms affecting the parvalbumin-expressing neurons in CHL1-/- mice remain elusive, we can assume that different interaction partners of CHL1 could play a role [42].” (lines 521-524)

It would be important to examine markers of cell death (e.g., apoptosis) or increased microglial phagocytic activity around these interneurons.

R: Previously, we had observed minimal apoptotic cell death in injured spinal cords. Therefore, we assume that most neurons die via necrotic cell death, followed by activation of microglia/macrophages, which are increased in numbers remote to the injury site. We now emphasize this in the Discussion: “The increase in numbers of microglia/macrophages remote from the lesion site likely refers to their roles in eliminating neurons, which undergo excitotoxic necrotic cell death and synaptic remodeling [6,8].” (lines 484-487)

While the study focused on parvalbumin-positive interneurons, it would also be important to analyze other interneuron subtypes (e.g., calbindin-positive or GABAergic populations).

R: We tried immunostainings with several other interneuronal markers (GAD67; calbindin), but none of them gave a staining pattern reliable enough for quantification. We now acknowledge this weakness of our study in the Discussion: “We tried to stain spinal cords with two other interneuronal markers (GAD67, calbindin), but the staining quality was not good enough for reliable quantifications.” (lines 513-515)

Regarding oligodendrocytes, did the authors assess myelin integrity using myelin-specific stains (Luxol Fast Blue or MBP ihc)?

R: We did not perform myelin-specific immunostaining. The reason for this was that there was no difference in the number of oligodendrocytes between the genotypes. Thus, we assumed that demyelination/remyelination after injury progresses at a similar pace in CHL1+/+ and CHL1-/- mice. We now acknowledge this as a weakness of our study in the Discussion: “Since the number of CNP-expressing oligodendrocytes was similar in both genotypes, we did not perform a more detailed analysis of remyelination after injury using myelin-specific markers, such as myelin basic protein. We acknowledge that this is a weakness of our study.” (lines 480-483)

Did the authors assess functional outcomes, such as locomotor performance (e.g., Basso Mouse Scale or other behavioral tests, f.e. video-based kinematic analysis)?

R: We now added data for functional outcomes in the new Figure 1.

Minor points:

“Among the cell adhesion molecules of interest for regeneration is the close homolog of adhesion molecule L1 (CHL1), which was shown to be upregulated by astrocytes at the lesion site after spinal cord injury [8]. This unexpected increase was considered to hamper regeneration through homophilic CHL1-CHL1 interactions between axons and astrocytes, leading to better recovery after injury in female, but not in male CHL1-deficient (CHL1-/-70) mice [8,9].” This section is somewhat difficult to follow due to its complex sentence structure and dense information. Rephrasing it for clarity and simplicity would greatly improve readability and help convey the intended message more effectively.

R: We have rephrased this statement in more sentences: “This upregulation of CHL1 expression was considered to hamper regeneration through homophilic CHL1-CHL1 interactions between axons and astrocytes, leading to better post-injury recovery in female, but not male CHL1−/− mice [8,9]. (lines 68-71)

The quality of microphotographs of immunodetection, particularly those of high power, needs substantial improvement. Please add better images to the manuscript and also quantify your staining as it is state-of-the-art.

R: We now show new figures and added more high-power images.

Round 2

Reviewer 2 Report

Comments and Suggestions for Authors

The authors have addressed my previous comments, which has improved the overall quality and clarity of the manuscript.